# *Weizmannia coagulans* BC179 Alleviates Post-Alcohol Discomfort May via Taurine-Related Metabolism and Antioxidant Regulation: A Randomized, Double-Blind, Placebo-Controlled Trial

**DOI:** 10.3390/antiox14091038

**Published:** 2025-08-23

**Authors:** Mengyao Duan, Ying Wu, Jie Zhang, Saman Azeem, Yao Dong, Zhonghui Gai, Jianguo Zhu, Shuguang Fang, Shaobin Gu

**Affiliations:** 1College of Food and Bioengineering, Henan University of Science and Technology, Luoyang 471000, China; mengyaoduan@stu.haust.edu.cn (M.D.); yingwu@haust.edu.cn (Y.W.); zhangjie@haust.edu.cn (J.Z.); samanazeem850@gmail.com (S.A.); 2Henan Engineering Research Center of Food Material, Henan University of Science and Technology, Luoyang 471023, China; 3Henan Engineering Research Center of Food Microbiology, Luoyang 471000, China; 4Wecare Probiotics R&D Centers (WPC), Wecare Probiotics Co., Ltd., Suzhou 215200, China; yao.dong@wecare-bio.com (Y.D.); zhonghui.gai@wecare-bio.com (Z.G.); cai.zhu@wecare-bio.com (J.Z.); frank.fang@wecare-bio.com (S.F.)

**Keywords:** *Weizmannia coagulans* BC179, taurine and hypotaurine metabolism pathway, oxidative stress, ameliorate post-drinking discomfort

## Abstract

Excessive alcohol consumption is associated with various health complications, including liver damage and systemic inflammation. Probiotic interventions have emerged as promising strategies to mitigate alcohol-induced harm, yet their mechanisms of action remain incompletely understood. This randomized, double-blind, placebo-controlled clinical trial aimed to evaluate the protective effects of *Weizmannia coagulans* BC179 in chronic alcohol consumers. Seventy participants with a history of long-term alcohol intake were randomly assigned to receive either BC179 (3 g/day, 1 × 10^10^ CFU) or a placebo for a 30-day intervention period. Following alcohol ingestion, dynamic monitoring of blood alcohol concentration (BAC), inflammatory and oxidative stress biomarkers, and serum metabolomic profiles was conducted. BC179 supplementation significantly reduced BAC and enhanced the activities of alcohol dehydrogenase (ADH) and aldehyde dehydrogenase (ALDH), while decreasing levels of alkaline phosphatase (ALP), high-sensitivity C-reactive protein (hs-CRP), tumor necrosis factor-α (TNF-α), and interleukin-6 (IL-6). Conversely, the anti-inflammatory cytokine interleukin-10 (IL-10), superoxide dismutase (SOD), and glutathione (GSH) were significantly upregulated. Levels of cytochrome P4502E1 (CYP2E1) and malondialdehyde (MDA) were also markedly reduced. Metabolomic analysis revealed significant modulation of taurine and hypotaurine metabolism, as well as downregulation of caffeine-related pathways. Collectively, these findings indicate that *W. coagulans* BC179 alleviates alcohol-induced discomfort by enhancing alcohol metabolism, attenuating inflammation, reducing oxidative stress, and modulating key metabolic pathways. This probiotic strain may represent a promising adjunctive strategy for managing alcohol-related health issues.

## 1. Introduction

Excessive alcohol consumption is a major global health challenge, driving a spectrum of liver injury from steatosis and alcoholic hepatitis to cirrhosis and liver failure [1,2]. In the liver, ethanol is oxidized by alcohol dehydrogenase (ADH) to acetaldehyde and subsequently by aldehyde dehydrogenase (ALDH) to acetate; with chronic intake, the microsomal ethanol-oxidizing system (largely CYP2E1) is induced and contributes substantially to acetaldehyde and reactive oxygen species (ROS) generation [3,4]. Acetaldehyde forms protein/DNA adducts, while CYP2E1-derived superoxide and hydrogen peroxide, together with an elevated NADH/NAD^+^ ratio, impair mitochondrial function, inhibit β-oxidation, and drive lipid peroxidation (e.g., MDA/4-HNE). These events activate Kupffer cells and downstream TLR4/NF-κB signaling, reduce antioxidant defenses (decreased activities of superoxide dismutase, catalase, and glutathione peroxidase with glutathione depletion), and culminate in hepatocellular injury and hepatic/systemic oxidative stress [5,6,7].

Beyond intrahepatic mechanisms, alcohol disrupts the gut-liver axis. Both acute and chronic drinking deplete commensals (e.g., *Bifidobacterium*, *Lactobacillus*) and enrich endotoxin-producing pathobionts (e.g., *Escherichia coli*, *Enterococcus*), increasing intestinal permeability and facilitating translocation of lipopolysaccharide (LPS) to the portal circulation. LPS augments hepatic inflammation via TLR4/NF-κB and acts synergistically with CYP2E1-derived oxidative stress to accelerate disease progression [8]. While over-the-counter remedies for post-alcohol symptoms are widely used, their efficacy is inconsistent, underscoring the need for interventions that target barrier dysfunction, dysbiosis, and oxidative stress.

*Weizmannia coagulans* BC179 (formerly *Bacillus coagulans*) is a spore-forming lactic acid-producing bacterium whose endospores confer high tolerance to heat, gastric acidity, and bile, favoring survival through the upper gastrointestinal tract. Although functional effects are inherently strain-dependent, convergent preclinical evidence across other strains of *W. coagulans* indicates species-level capabilities relevant to alcohol-related physiology: modulation of gut microbial communities with enrichment of beneficial genera (e.g., *Lactobacillus*, *Bifidobacterium*) and suppression of opportunists (e.g., *Escherichia coli*, Enterococcus), reinforcement of intestinal barrier integrity, reduction of circulating lipopolysaccharide (LPS), and attenuation of ethanol- or inflammation-associated hepatic injury in animal models [9,10]. Taken together, these data provide a biologically plausible species-level rationale for evaluating BC179 as an adjunct for alcohol-related health management. We therefore hypothesize that BC179 may act along the gut-liver axis to fortify barrier function, limit endotoxin translocation, and temper oxidative and inflammatory signaling; however, in view of known inter-strain heterogeneity, these putative benefits require strain-specific clinical confirmation, which is the focus of the present trial.

However, despite promising preclinical evidence, the efficacy and underlying mechanisms of BC179 in humans remain unvalidated. In particular, its effects on key biomarkers of alcohol metabolism, systemic inflammation, redox homeostasis, and host serum metabolomic signatures have not been systematically assessed in clinical populations. Therefore, this study aimed to evaluate the effects of BC179 supplementation in habitual alcohol consumers through a randomized, double-blind, placebo-controlled clinical trial. We hypothesized that BC179 would mitigate post-alcohol discomfort and associated physiological disruptions by enhancing alcohol metabolism, reducing inflammatory and oxidative stress markers, and modulating serum metabolic pathways. This work may offer novel insights into the clinical utility of probiotics for alcohol-related health management.

## 2. Materials and Methods

### 2.1. Research Object

This study was conducted in accordance with the ethical principles of the World Medical Association’s Declaration of Helsinki [11], and approved by the Ethics Committee of the First Affiliated Hospital of Henan University of Science and Technology. The clinical trial registration number is NCT06899620. The clinical trial was registered at ClinicalTrials.gov prior to participant enrolment. All investigators were trained in Good Clinical Practice (GCP). A total of healthy adult volunteers (*n* = 100), aged 18 to 65 years, were enrolled between July and November 2024. Participants were eligible for inclusion if they were aged 18–65 years, had a documented history of long-term alcohol consumption (≥20 g of ethanol per day for at least one year, calculated as: volume (mL) × alcohol content (%) × 0.8), and had provided written informed consent with a willingness to comply with study procedures. Individuals were excluded if they had a history of chronic liver, gastrointestinal, or systemic metabolic disorders; had used products with similar functions to the investigational formulation within the past two weeks; had taken antibiotics, laxatives, or dietary supplements in the past four weeks; had a documented history of hypertension or diabetes; had severe allergies or immune deficiencies; or were pregnant, lactating, or planning pregnancy without effective contraception during the study period.

#### 2.1.1. Pilot Crossover Study

A randomized, double-blind, two-period crossover trial was conducted to preliminarily evaluate the efficacy of BC179 and to refine the experimental protocol for the subsequent main study. The experimental process is shown in Figure 1. Thirty habitual drinkers were randomly assigned (1:1) to BC179 group (BC179→placebo) or placebo group (placebo→BC179). Each intervention period lasted 7 days and was separated by a 14-day washout period to eliminate residual effects and ensure data independence. During each intervention phase, participants received a daily sachet of either 3 g probiotic powder (containing 10 billion CFU BC179) or placebo (3 g of Dextrin). Both products were identical in appearance, shape, color, flavor, sweetness, packaging, and excipients to maintain their regular diet and lifestyle. On day 7 of each intervention phase, after consuming a standardized light meal, participants ingested Dukang liquor (52% alcohol by volume) at a dose of 0.3 g ethanol per kilogram of body weight. Breath alcohol concentrations and venous blood samples were collected at 0, 15, 30, 60, 120, 180, 240, 300, and 360-min post-ingestion. Participants also completed the Number Connection Test (NCT-A) immediately before and 60 min after alcohol consumption to assess cognitive-motor performance. The session lasted at least 120 min and continued until two consecutive breath tests returned negative. BAC, ADH, and ALDH levels were quantified using commercially available assay kits (Yishijiu Biotechnology Co., Ltd., Lianyungang, China) in accordance with the manufacturer’s instructions. The primary purpose of this pilot was to estimate the preliminary effect size of BC179 on alcohol metabolism and cognitive recovery, and to assess the feasibility of test indicators for the full-scale trial.

#### 2.1.2. Main Randomized Controlled Trial

Based on the findings of the pilot trial, a randomized, double-blind, placebo-controlled clinical study was conducted, enrolling a total of 70 eligible participants. The experimental process is shown in Figure 1. Subjects were randomly allocated into two parallel arms using a computer-generated randomization schedule: the probiotic group (*n* = 35) received BC179 (3 g/day, containing 10 billion CFU), while the placebo group (*n* = 35) received a matching placebo (3 g/day of Dextrin). Both interventions were provided in identical sachets, matched for appearance, taste, packaging, and excipient content to ensure blinding. The study duration was 60 days. Baseline data were collected on Day 0, and follow-up assessments were conducted on Day 45 ± 15. During the follow-up visit, participants were asked to consume a standardized alcohol challenge under controlled conditions. Clinical symptoms, biochemical parameters, and adverse events within 24 h post-ingestion were recorded and analyzed. Participants were instructed to consume the designated product daily in accordance with the recommended dietary ratio (50–60% carbohydrates, 15–20% protein, 20–30% fat), and to keep a research diary to record adherence and any adverse reactions. A total of 60 participants completed all study procedures, and data from these individuals were included in the final analysis.

### 2.2. Hangover Symptom Questionnaire

A self-administered hangover questionnaire was distributed using the questionnaire star platform, with a link and QR code disseminated via the social media platform WeChat. The questionnaire collected demographic data, average alcohol consumption, and the presence of hangover symptoms such as dizziness and nausea following alcohol intake. Questionnaire completion progress was monitored in real-time through the backend system to ensure data integrity. Only fully completed and valid questionnaires were included in the final analysis. The total hangover symptom score ranged from 0 to 36, based on the sum of individual symptom scores. Severity was categorized into five levels: No hangover symptoms (0), Mild symptoms (1–8), Moderate symptoms (9–15), Moderate-to-severe symptoms (16–23), Severe hangover symptoms (24–36) [12]. A copy of the full questionnaire is provided in Appendix A.

### 2.3. Blood Sample Collection and Determination

During the second month of the intervention period, blood samples were collected at Henan University of Science and Technology Hospital using clinical standard measurements on the day after the participants had consumed alcohol. Morning blood samples were collected after at least 10 h of fasting for index analysis. The serum samples were centrifuged and stored at −80 °C for index determination and analysis. The following indicators were detected using commercial kits according to the manufacturer’s protocol (He Peng (Shanghai) Biotechnology Co., Ltd., No. 3688, Tingwei Road, Caojing Town, Jinshan District, Shanghai, China): ADH, (HP-E10895), ALDH, (HP-E10896), hs-CRP, (HP-E11183), TNF-α, (HP-E10110), IL-6, (HP-E10140), IL-10, (HP-E10155), SOD, (HP-E11086), GSH, (HP-E2064), MDA, (HP-E10376), and LPS, (HP-E10787).

### 2.4. Serum Metabolomic Analysis

Sample Preparation: Samples preserved at −80 °C were allowed to defrost at room temperature. Eighty microliters of each sample were aliquoted into a 1.5 mL Eppendorf tube, followed by the addition of 4 μL L-2-chlorophenylalanine (0.06 mg/mL in methanol) as an internal standard. The mixture was vortexed for 10 s, after which 320 μL of a chilled methanol-acetonitrile mixture (2:1, *v*/*v*) was added. The solution was vortex-mixed for 1 min, then subjected to ultrasonic extraction for 10 min using an ice-water bath. Samples were kept at −40 °C overnight, followed by centrifugation at 12,000 rpm for 20 min at 4 °C. A 150 μL aliquot of the supernatant was transferred to an LC-MS vial with a glass insert for analysis. Quality control (QC) samples were prepared by pooling aliquots from all individual samples.

Instrumental Analysis: Analyses were performed using a Waters ACQUITY UPLC I-Class plus system coupled with a Thermo QE HF UHPLC-HRMS (ultra-high-performance liquid chromatography-tandem high-resolution mass spectrometer). Separation was achieved on an ACQUITY UPLC HSS T3 column (1.8 μm, 2.1 mm) under both positive and negative electrospray ionization modes. The mobile phase comprised (A) 0.1% formic acid in water (*v*/*v*) and (B) 0.1% formic acid in acetonitrile (*v*/*v*), with a gradient elution program as follows: 0.01–2 min: 5% 2–4 min: 5%→30% 4–8 min: 30%→50% 8–10 min: 50%→80% 10–14 min: 80%→100% 14–15 min: 100% 15–16 min: re-equilibration at 5%. The flow rate was 0.35 mL/min, the column temperature was set to 45 °C, and samples were stored at 10 °C in the autosampler. The injection volume was 4 μL. Mass spectrometry parameters were set as follows: mass range *m*/*z* 100–1000; resolution 70,000 for full MS scans and 17,500 for HCD MS/MS scans; collision energies 10, 20, and 40 eV; spray voltage 3800 V (+) and 3200 V (−); sheath gas flow 35 arb. units; auxiliary gas flow 8 arb. units; capillary temperature 320 °C; auxiliary gas heater temperature 350 °C; and S-lens RF level 50. Metabolomic data analysis was conducted by Shanghai Luming Biological Technology (Shanghai, China).

### 2.5. Statistical Analysis

Statistical analyses were conducted using SPSS 25.0 software. Experimental results are expressed as mean ± standard deviation with a 95% confidence interval. Significance was defined as: (Non-significant: *p* > 0.05; Significant: 0.01 ≤ *p* < 0.05; Highly significant: 0.001 ≤ *p* < 0.01). For comparisons between groups, independent sample *t*-tests were employed when data satisfied the assumptions of normal distribution and variance homogeneity. In cases where these assumptions were not met, the Mann–Whitney U test was utilized. Visualizations were created using GraphPad Prism 10.1.2.

Raw LC-MS data were analyzed using Progenesis QI V2.3 (Nonlinear Dynamics, Newcastle, UK) to perform baseline filtering, peak identification, integration, retention time correction, peak alignment, and normalization. Compounds were identified by matching precise *m*/*z* values, MS/MS fragments, and isotopic distributions against the Human Metabolome Database (HMDB), Lipidmaps (V2.3), Metlin, and in-house databases. An integrated data matrix was generated by combining positive and negative ion datasets. The combined dataset was loaded into R for principal component analysis (PCA) to evaluate sample distribution and analytical stability. Orthogonal partial least-squares discriminant analysis (OPLS-DA) and partial least-squares discriminant analysis (PLS-DA) were applied to screen for differentially expressed metabolites between groups. To prevent overfitting, model validity was assessed via 7-fold cross-validation and 200 iterations of Response Permutation Testing (RPT).

Variable Importance in Projection (VIP) scores from OPLS-DA were applied to rank variables according to their contribution to group discrimination. Inter-group metabolic differences were validated using two-tailed Student’s *t*-tests for statistical significance. Differentially expressed metabolites were identified using criteria of VIP > 1.0 and *p* < 0.05, followed by KEGG (http://www.genome.jp/kegg/) pathway enrichment analysis (accessed on 1 February 2024).

## 3. Results

### 3.1. Pre-Experiment Results

#### 3.1.1. Baseline Characteristics

The baseline demographic and anthropometric characteristics of the participants in BC179 and placebo groups are summarized in Table 1. The mean ages of participants were 30.40 ± 13.16 years in the BC179 group and 31.93 ± 12.58 years in the placebo group. The mean BMI was 23.67 ± 2.83 kg/m^2^ and 22.81 ± 2.54 kg/m^2^, respectively. No statistically significant differences were found between the groups in terms of age, BMI, or sex distribution (*p* > 0.05), confirming comparability at baseline.

#### 3.1.2. Serum ADH and ALDH Activity Within 4 h Post-Alcohol Intake

As shown in Figure 2A, B, serum ADH levels in the BC179 group were significantly higher than those in the placebo group at 30 min post-drinking (*p* < 0.05), with no significant difference observed at 1 h. Figure 2B illustrates changes in ALDH activity. ALDH levels were significantly elevated in the BC179 group compared to the placebo group at 15 min (*p* < 0.05), 30 min (*p* < 0.001), and 1 h (*p* < 0.001) following alcohol consumption. These results suggest that BC179 may accelerate ethanol and acetaldehyde metabolism during the early post-drinking phase.

#### 3.1.3. Changes in Alcohol Level Within 4 h After Drinking

Figure 2C shows that breath alcohol concentration was significantly lower in the BC179 group than in the placebo group at 15 min (*p* < 0.01) and 30 min (*p* < 0.05) after drinking. Notably, 100% of subjects in the BC179 group achieved a breath alcohol concentration of zero by 180 min, whereas the placebo group did not reach this threshold until 240 min. Similarly, Figure 2D demonstrates a faster reduction in blood alcohol concentration in the BC179 group. At 15 min and 30 min post-drinking, blood alcohol levels were significantly lower than in the placebo group (*p* < 0.05 and *p* < 0.01, respectively). By 120 min, 79.4% of participants in the BC179 group had undetectable blood alcohol levels, compared to 58.7% in the placebo group.

#### 3.1.4. Number Connection Test (NCT-A)

The Number Connection Test was used to assess psychomotor performance and cognitive processing speed. Participants were instructed to sequentially connect numbers from 1 to 25 without lifting the pen, and completion time was recorded in seconds. Errors were corrected in real time without pausing the timer. Figure 2E shows no significant change in NCT-A performance in the placebo group after intervention. However, participants in the BC179 group completed the test significantly faster at 60 min post-alcohol intake compared to the placebo group (*p* < 0.05), indicating a positive effect of BC179 on short-term cognitive function following alcohol consumption.

### 3.2. Formal Experimental Results

The baseline characteristics of participants are presented in Appendix A, including mean age, sex distribution, and BMI. The placebo group had a mean age of 43.60 ± 11.31 years, while the BC179 group showed 42.87 ± 11.15 years. All participants in the placebo group were male, compared to approximately 96.6% in the BC179 group. BMI values were 24.24 ± 2.89 kg/m^2^ for placebo and 24.87 ± 2.68 kg/m^2^ for BC179, with no significant intergroup differences. No significant group differences were observed in other measures, including renal and liver function, body composition, and biochemical parameters.

#### 3.2.1. Hangover Questionnaire

The hangover questionnaire can provide quantitative indicators of the severity of alcohol hangovers in a simple and reliable way. Under this questionnaire system, the score is positively correlated with the intensity of discomfort after drinking alcohol, that is, the higher the score, the more intense the discomfort experienced by the subjects after drinking alcohol. At the time point of taking the product for (45 ± 15) days, the hangover scores of the two groups of subjects were analyzed. As shown in Figure 3A, the placebo group exhibited no significant change in hangover scores between the pre- and post-intervention periods. In contrast, the BC179 group showed a 7.70-point reduction in hangover scores post-intervention, demonstrating a highly significant downward trend (*p* < 0.001).

#### 3.2.2. Alcohol Metabolizing Enzymes and Blood Alcohol Concentration

ADH facilitates the oxidation of ethanol to acetaldehyde, whereas ALDH catalyzes the subsequent conversion of acetaldehyde to acetic acid, which is ultimately metabolized into carbon dioxide and water. Following a minimum 30-day intervention, Figure 3B,C illustrate that the BC179 group exhibited significantly elevated post-alcohol ADH and ALDH levels compared to the placebo group (*p* < 0.05). Specifically, mean ADH and ALDH concentrations increased by 5.00 ng/mL and 1.8 ng/mL, respectively. Concurrently, Figure 3D demonstrates that after at least 30 days of intervention, the BC179 group showed a significant 0.57 μmol/mL reduction in blood alcohol concentration relative to the placebo group (*p* < 0.05).

#### 3.2.3. Serum Alkaline Phosphatase and Intestinal Barrier

From Figure 3E,F, the BC179 group exhibited a significant post-alcohol decrease in ALP levels (*p* < 0.001), with a 5.55 ng/mL reduction compared to the placebo group. Additionally, the study revealed that blood LPS levels in the BC179 group were significantly lower than those in the placebo group by 126.99 EU/L at 24 h post-alcohol consumption (*p* < 0.001).

#### 3.2.4. Inflammatory Factors

As an important acute phase protein, hs-CRP is an extremely sensitive marker of inflammatory response. As shown in Figure 4, through rigorous experimental comparison, compared with the placebo group, the hs-CRP content of the BC179 group decreased by 3.28 mg/L after drinking for 24 h, showing a very significant decreasing trend (*p* < 0.001). After drinking, macrophages of the immune system release pro-inflammatory factors such as TNF-α and IL-6. Compared with the placebo group, the BC179 group showed a significant decline in the content of pro-inflammatory factors TNF-α and IL-6 24 h after drinking. At the same time, the content of anti-inflammatory factor IL-10, which played a role in alleviating inflammation, showed a very significant increase (*p* < 0.001), with an average increase of 9.63 pg/mL.

#### 3.2.5. Oxidative Stress

The SOD levels of the placebo and BC179 groups were checked before and after a 60-day treatment, and the results are shown in Figure 5A. After 30 days of BC179 intervention, the BC179 group exhibited significantly higher SOD levels than the placebo group (*p* < 0.01), with a mean increase of 11.75 U/mL. Concurrently, as a key endogenous antioxidant, GSH plays a critical role in maintaining redox homeostasis. The GSH levels in both groups were measured before and after the intervention (Figure 5B), showing a significant increase in the BC179 group after the intervention (*p* < 0.05), with an average rise of 0.14 μmol/mL compared to the placebo group.

CYP2E1 is known to generate ROS during alcohol metabolism, contributing to oxidative stress. As shown in Figure 5C, the BC179 group had a significant drop in CYP2E1 levels after the intervention (*p* < 0.05), with an average decrease of 3.1 ng/mL compared to the placebo group. Additionally, MDA, a biomarker of lipid peroxidation, was assessed (Figure 5D). The BC179 group showed a significant drop in MDA levels after the treatment (*p* < 0.001), with an average decrease of 0.10 μmol/mL compared to the placebo group, suggesting less oxidative damage.

#### 3.2.6. Serum Metabolomics

##### Differential Metabolites

The present study focused on the analysis of the modulation of human metabolites by BC179 versus placebo after drinking alcohol. OPLS-DA was used to identify relevant biomarkers between the BC179 group and the placebo group after alcohol consumption. Sample points were distributed on both sides of the space, indicating significant separation between the two groups and differences that could be analyzed in subsequent experiments (variable effect on prediction, VIP > 1). Separation was observed in permutation plots of samples from the BC179 and placebo groups (R^2^ = 0.0773, Q^2^ = −0.508) (Figure 6A,B). The serum metabolic profile was presented by PCA analysis, and the cumulative proportion of principal components 1, 2, and 3 accounted for 29.83% of the total components. Statistical analysis revealed clear differences in metabolic profiles between the BC179 and placebo groups and significant differences in metabolite regulation, as shown in Figure 6C,D. A total of 235 serum metabolites were significantly upregulated and 475 serum metabolites were significantly downregulated in the BC179 group compared with the placebo group, see Figure 6D.

Further analysis was conducted on the top ten differential metabolites that were significantly upregulated and downregulated between the BC179 group and the placebo group. It was found that they were classified into a total of 10 categories: heterocyclic compounds, ceramide lipids, flavonoids and their derivatives, anthocyanins, peptides, quinones, alcohols, esters, glycosides, and terpenoids. Among these types of substances, lipid substances classified as ceramides all showed significant up-modulation of Cer (D20:1/6 Keto-PGF1Alpha), (6-keto-PGF1alpha), Cer (d18:2/PGF2alpha), and Cer (d16:1/22:6-2OH), N-arachidonoyl-dopamine-d8 in the BC179 group see Figure 7A. However, quinones such as SCHEMBL2081596 and Plastoquinone 3 were significantly downregulated in the BC179 group as shown in Figure 7B, C. It can be seen from this that BC179 intervention has a significant effect on serum metabolism; compared with the placebo intervention, BC179 significantly regulated 710 serum metabolites. Among them, the levels of key metabolites related to taurine metabolism, such as 5-L-glutamyl-taurine, Taurocholic acid and Taurochenodeoxycholic acid, have undergone significant changes.

##### Effects of BC179 on Metabolic Pathways

Key pathways involved in 20 metabolic processes were identified by serum metabolites (VIP > 1, *p* < 0.05) between the placebo and BC179 groups. As shown in Figure 8, seven metabolic pathways (*p* < 0.1) showed changes after the BC179 intervention as compared with the placebo intervention, which plays a key role in the body’s metabolic processes. Its physiological function covers multiple aspects, such as antioxidant adjusting the cell osmotic pressure taurine and hypotaurine metabolism pathways, and mainly depends on the cytochrome P450 enzyme system caffeine metabolism of metabolic pathways, which changed significantly. In addition, there are alterations in the steroid hormone biosynthesis pathway, which produces glucocorticoids that regulate carbohydrate metabolism, mineral balance, and immune responses to help cope with stress. It has immunosuppressive effects, helps to maintain the stability of the internal environment, and plays an important role in a variety of metabolic processes in the organism. It is a key enzyme in balancing the intracellular glucose metabolism and detoxification. Energy metabolism is of great significance since more obvious changes also appeared in the Pentose and glucuronate interconversion metabolic pathways.

##### Correlation Analysis

Correlation analysis was performed between key serum metabolites and clinical indicators, and metabolites with significant changes in metabolic pathways were used to reveal the association between metabolic markers and apparent indicators of participants. Figure 9 directly shows that the three metabolites involved in caffeine metabolism (Theophylline, Theobromine, and caffeine) exhibit a significant positive correlation with P450E1. Meanwhile, Taurocholic acid and Taurochenodeoxycholic acid, which participate in the taurine and hypotaurine pathway, display a significant positive correlation with P450E1, SOD, and IL-10, and a significant negative correlation with TNF-α. Among these, Taurocholic acid also shows a marked positive correlation with GSH and ADH. The results suggest that these serum metabolites may serve as key markers influencing disease phenotypes through the caffeine or taurine and hypotaurine metabolic pathways.

## 4. Discussion

Against the backdrop of the diverse modern society, alcohol consumption has become a common social activity. However, the accompanying post-drinking discomfort, including hangover symptoms, physiological stress, and fatty liver-related diseases, has attracted increasing attention from the public as well as the health and scientific research communities [13,14]. Conventional intervention strategies have demonstrated limited efficacy in mitigating these symptoms. In contrast, probiotics, due to their ability to modulate the host’s gut microbiota and maintain a microecological balance, have emerged as promising candidates for biological regulation of alcohol-related discomfort [15]. The present study focused on evaluating the effects of *Weizmannia coagulans* BC179 in individuals with chronic alcohol consumption, and the results demonstrated that BC179 significantly alleviated post-drinking discomfort in this population.

Ethanol is initially absorbed in the stomach and rapidly enters the small intestine, where it is efficiently absorbed into the bloodstream [15]. It is then transported to the liver, where metabolic breakdown begins. ADH is a key enzyme responsible for catalyzing the first step of ethanol oxidation, converting ethanol into acetaldehyde by removing two hydrogen atoms from the molecule [16,17]. Acetaldehyde, a toxic intermediate, is subsequently oxidized to non-toxic acetic acid by ALDH, which is then further metabolized and eliminated from the body [18,19]. In this study, participants who received BC179 supplementation exhibited significantly elevated serum levels of both ADH and ALDH compared to the placebo group. Concurrently, BAC levels were significantly reduced. These findings suggest that BC179 enhances the enzymatic capacity for alcohol metabolism, thereby accelerating ethanol clearance and lowering systemic alcohol levels.

In addition to promoting alcohol metabolism, BC179 intervention also improved hepatic function and intestinal barrier integrity. Specifically, participants in the BC179 group showed significantly lower serum levels of ALP and LPS. ALP is a hydrolase enzyme expressed in various tissues, particularly the liver and bones, and its elevated serum levels have been associated with hepatic inflammation and oxidative stress [20]. ALP is not only considered a biomarker of liver injury [21] but also plays an active role in promoting inflammation by activating the NF-κB signaling pathway, which induces the expression of pro-inflammatory cytokines such as TNF-α and IL-6 [22]. The observed decrease in ALP following BC179 intervention suggests a protective effect on hepatic and inflammatory responses.

Alcohol-induced disruption of the gut barrier increases intestinal permeability, permitting the translocation of bacteria and their endotoxins, most notably LPS, into the portal circulation [23]. Circulating LPS activates macrophages, stimulates release of pro-inflammatory cytokines (TNF-α, IL-6), and triggers systemic oxidative stress via reactive ROS generation [24]. Long-term high LPS levels can lead to intestinal leakage, organ dysfunction, and insulin resistance [25]. In the present study, BC179 supplementation markedly lowered serum LPS together with TNF-α and IL-6, while significantly increasing the anti-inflammatory cytokine IL-10 and reducing high-sensitivity CRP. Downregulation of these pro-inflammatory mediators can interrupt NF-κB-dependent pathways that drive steatosis and hepatic inflammation, thereby attenuating ALD progression [26,27,28].

In cases of frequent and excessive alcohol consumption, approximately 20% of alcohol undergoes acetaldehyde oxidation through the microsomal ethanol oxidation system (MEOS) [29], a process that mainly relies on cytochrome P4502E1 (CYP2E1) and is accompanied by the production of a large amount of ROS. These free radicals can trigger oxidative stress responses, causing damage to cells and tissues, and thereby aggravating discomfort after drinking [30,31,32]. Cytochrome P450 is a major metabolic enzyme in hepatocytes and plays a significant role in inflammatory responses and promoting cell apoptosis. It mainly undertakes 90% of the biotransformation and metabolic tasks of exogenous drugs (environmental toxins, anti-cancer drugs) and endogenous compounds (steroids, fatty acids, hormones) [33]. As one of the crucial free radical scavengers in the body, GSH shoulders the important responsibility of maintaining the oxidative balance. The activity of SOD reflects the antioxidant capacity of the body. After the BC179 treatment, the amounts of SOD and GSH went up a lot, while the level of P450E1, an important enzyme that makes free radicals, went down, and the amount of MDA, which leads to oxidative stress, also went down. These findings further underscore that BC179-mediated alleviation of acute alcoholic liver injury is tightly coupled to its antioxidant properties. Concomitantly, BC179 supplementation downregulates CYP2E1 while upregulating ADH, thereby mitigating alcohol metabolism-derived ROS production and bolstering hepatic protection. This is consistent with the study of Vetriselvan Subramaniyan et al. [34], that is, alcohol intake significantly induces CYP2E1 expression, leading to the production of a large amount of ROS, which in turn triggers lipid peroxidation, DNA damage, and hepatocyte apoptosis.

To further explore the potential mechanism of BC179 in alleviating alcohol and enhancing alcohol metabolism, serum metabolomics analysis was performed between the placebo group and the BC179 group. The results showed that the abundance of 710 serum metabolites showed significant changes after BC179 intervention. Among them, the abundance of 235 metabolites was upregulated and 475 metabolites were downregulated. Encyclopedia by KEGG enrichment analysis found that, after the supplementary BC179, taurine and hypotaurine metabolism rise significantly and caffeine metabolism pathways are significantly lower. As a substance with a variety of important physiological functions, taurine plays a key role in the metabolism of the body [35]. Its physiological functions include anti-oxidation, inhibition of inflammatory mediator’s release, regulation of cell osmotic pressure, and so on. In the process of detoxing, alcohol metabolism will promote the production of a large number of reactive oxygen species, which will cause oxidative stress damage and damage to important organs such as the liver [36]. Stella Baliou [37] has shown that taurine exerts protective effects in a variety of organs, such as the heart, liver, kidneys, and central nervous system, by reducing damage to these organs under oxidative stress conditions through antioxidant mechanisms. It is highly likely that taurine, with its antioxidant and anti-inflammatory properties, can effectively reduce the damage of alcohol to organs such as the liver, thus playing an indispensable role in the metabolic regulation process related to hangovers [38]. Caffeine is mainly metabolized in the liver by cytochrome P450 enzymes (such as CYP1A2) to produce a variety of bioactive metabolites, such as parra essence, theobromine, and theophylline, while alcohol metabolism is mainly carried out in the liver. Alcohol will slow down the clearance rate of caffeine and prolong the residence time of caffeine in the body. The decrease of P450E1 content in the experimental results also confirms this [39].

In addition, the metabolic pathways of steroid hormone biosynthesis, as well as Pentose and glucuronate interconversion metabolic pathways, also present a more obvious change, and the relevant results show that the key metabolites related to caffeine metabolism—theophylline, theobromine and caffeine—are significantly positively correlated with P450E1. Taurocholic acid and taurochenodeoxycholic acid are key metabolites involved in the metabolic pathways of taurine and sub-taurine, significantly positively correlated with P450E1, SOD, and IL-10, and significantly negatively correlated with TNF-α. Among them, taurocholic acid is also significantly positively correlated with GSH and ADH. These results fully indicate that BC179-mediated effects on relieving discomfort after drinking alcohol and enhancing alcohol metabolism are closely related to the changes of these metabolites and related metabolic pathways.

Combining the finding by You-Liang Hsieh et al. [40,41,42] that taurine can increase the levels of ADH and ALDH in patients with chronic alcoholism, along with the results of this study showing that taurine metabolic pathways are significantly enhanced and ADH activity is upregulated after BC179 intervention, we hypothesize that taurine may play a synergistic or mediating role in BC179-dependent enhancement of ADH activity. The potential mechanisms can be analyzed as follows: On one hand, taurine itself may be involved in the regulation of ADH expression through direct or indirect means. On the other hand, BC179 may indirectly promote the production and utilization of taurine by regulating gut microbiota metabolites, thereby synergistically enhancing ADH activity. In summary, taurine is likely not the sole factor directly involved in BC179-mediated regulation of ADH. Instead, through its antioxidant, anti-inflammatory, and metabolic regulatory properties, taurine forms a synergistic effect with BC179 to jointly promote the active expression of ADH, thereby accelerating ethanol clearance and alleviating post-drinking discomfort.

## 5. Conclusions

This study confirmed that *Weizmannia coagulans* BC179 enhances alcohol metabolism by upregulating ADH and ALDH, thereby accelerating ethanol clearance and lowering blood alcohol concentration. It also improves mucosal barrier function, reduces inflammatory markers (ALP, LPS, hs-CRP, TNF-α, and IL-6), and increases the anti-inflammatory cytokine IL-10. Additionally, BC179 boosts antioxidant capacity by modulating taurine and hypotaurine metabolism, increasing SOD and GSH levels, and decreasing P450E1 and MDA. After 30 days of intervention, BC179 significantly alleviated post-drinking discomfort in chronic drinkers, alongside reducing inflammation and oxidative stress. Serum metabolomics further supported these effects. These findings underscore the therapeutic potential of BC179, warranting further investigation into its mechanisms at the molecular and genetic levels to support broader clinical applications.

## Figures and Tables

**Figure 1 antioxidants-14-01038-f001:**
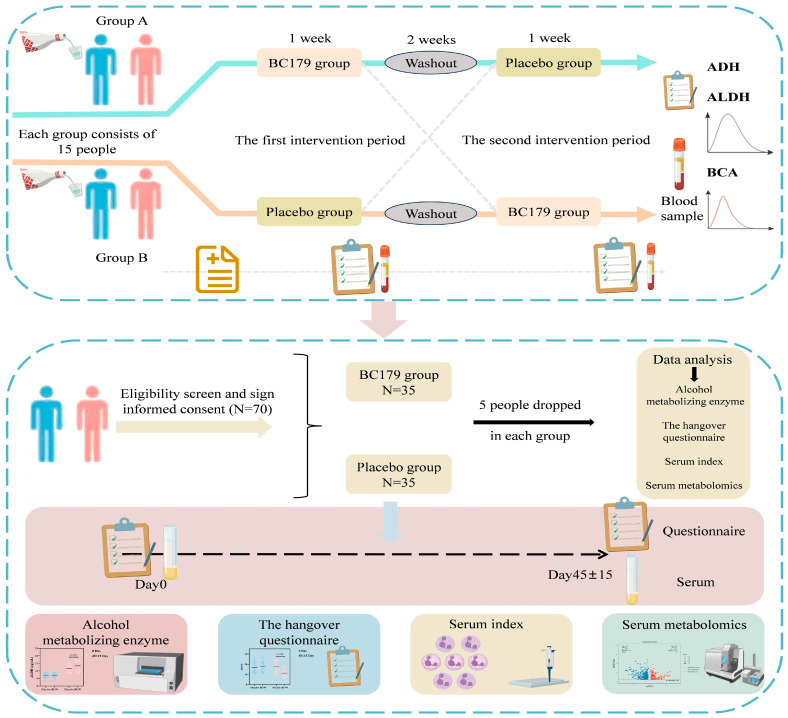
Flow diagram of the experimental study design.

**Figure 2 antioxidants-14-01038-f002:**
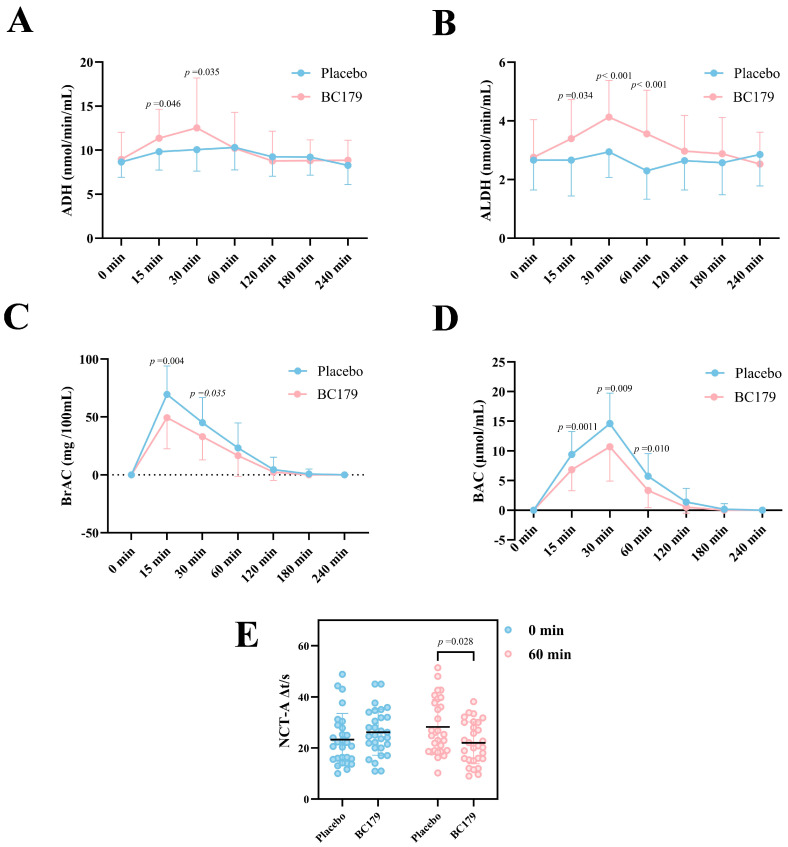
The contents of ADH, ALDH, BrAC, BAC, and NCT-A at different time points after drinking alcohol. (**A**) The changes in serum ADH activity. (**B**) The changes in serum ALDH activity. (**C**) changes in expiratory alcohol concentration. (**D**) The changes in serum BAC. (**E**) The changes in continuous digital testing.

**Figure 3 antioxidants-14-01038-f003:**
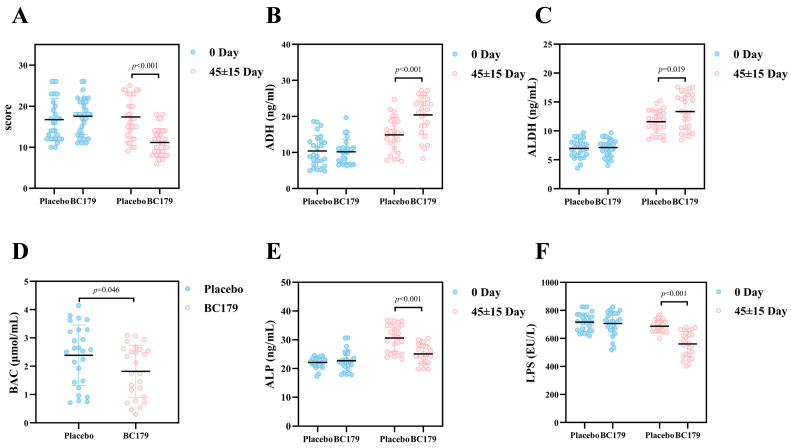
Changes of alcohol-related indicators 24 h after drinking in the placebo group and the BC179 group. (**A**) Changes in the hangover scores of the subjects after BC179 intervention. (**B**) The changes in serum ADH activity. (**C**) The changes in serum ALDH activity. (**D**) The changes in serum BAC. (**E**) The changes in serum ALP activity. (**F**) The changes in serum LPS.

**Figure 4 antioxidants-14-01038-f004:**
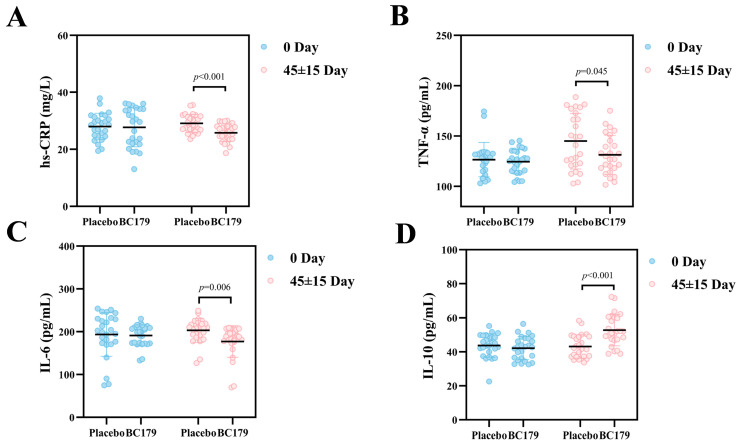
Changes of hs-CRP, TNF-α, IL-6, and IL-10 24 h after alcohol consumption in the placebo group and the BC179 group. (**A**) Changes in serum hs-CRP. (**B**) Changes in serum TNF-α. (**C**) Changes in serum IL-6. (**D**) Changes in serum IL-10.

**Figure 5 antioxidants-14-01038-f005:**
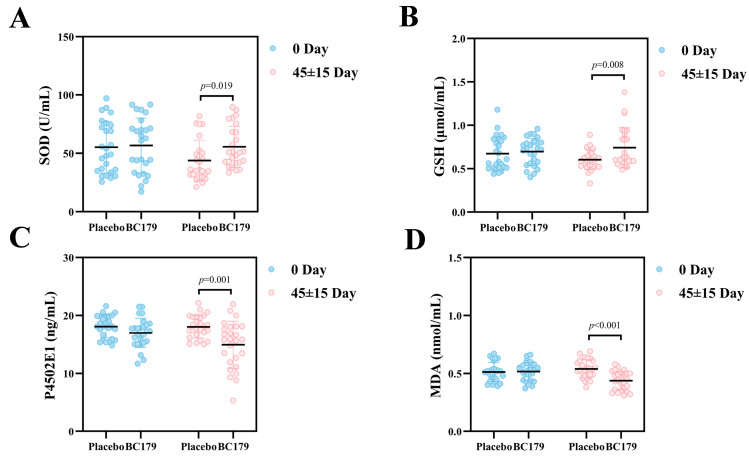
Changes in SOD, GSH, P4502E1, and MDA 24 h after drinking alcohol in subjects after BC179 intervention (45 ± 15). (**A**) Changes in serum SOD. (**B**) Changes in serum GSH. (**C**) Changes in serum P4502E1. (**D**) Changes in serum MDA.

**Figure 6 antioxidants-14-01038-f006:**
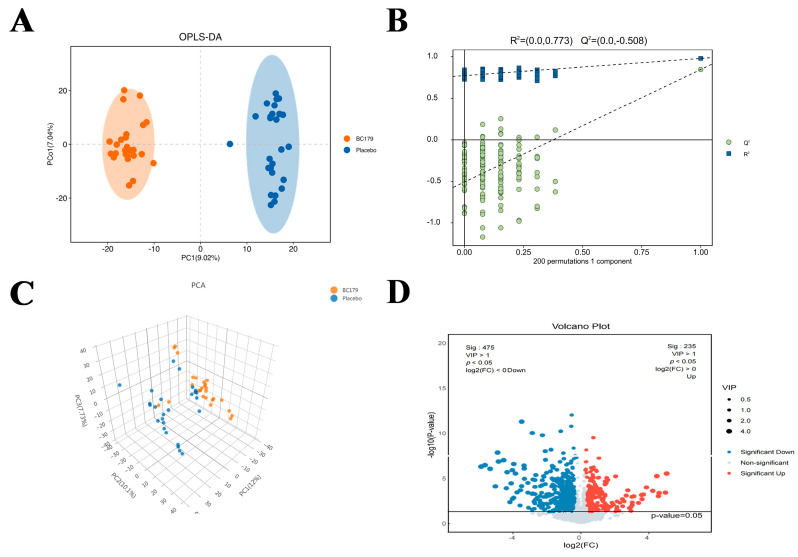
Effect of BC179 intervention (45 ± 15) on serum metabolic characteristics of subjects 24 h after alcohol consumption. (**A**) OPLS-DA permutation test. (**B**) permutation plot. (**C**) PCA score plot. (**D**) volcano plot.

**Figure 7 antioxidants-14-01038-f007:**
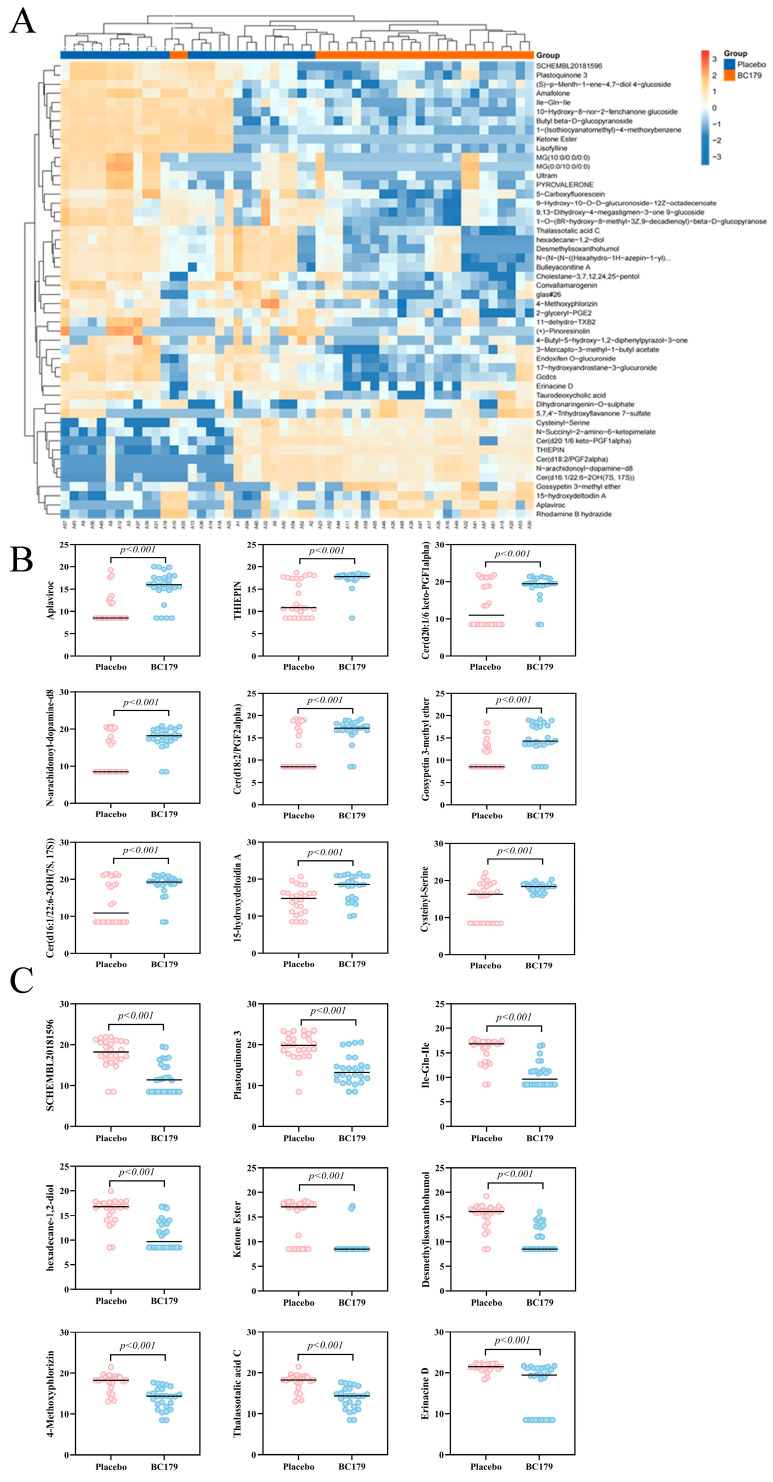
BC179 intervention (45 ± 15) differential metabolites in subjects. (**A**) Top 50 heat map of differential metabolites. (**B**) Upregulate the top nine relative abundances of metabolites. (**C**) Downregulate the top nine relative abundances of metabolites.

**Figure 8 antioxidants-14-01038-f008:**
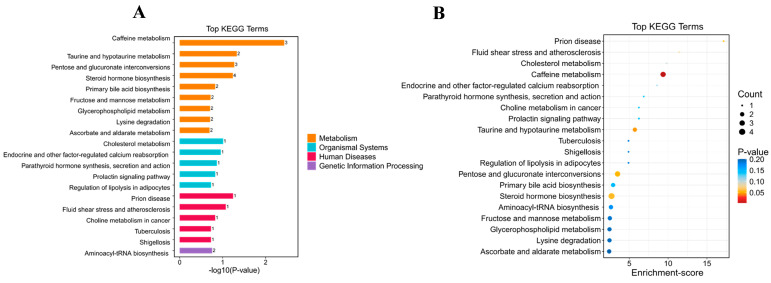
Effect of BC179 intervention (45 ± 15) on serum metabolic pathways of subjects 24 h after alcohol consumption. (**A**) KEGG Level 3 level distribution map of differential metabolites. (**B**) KEGG analysis bubble map.

**Figure 9 antioxidants-14-01038-f009:**
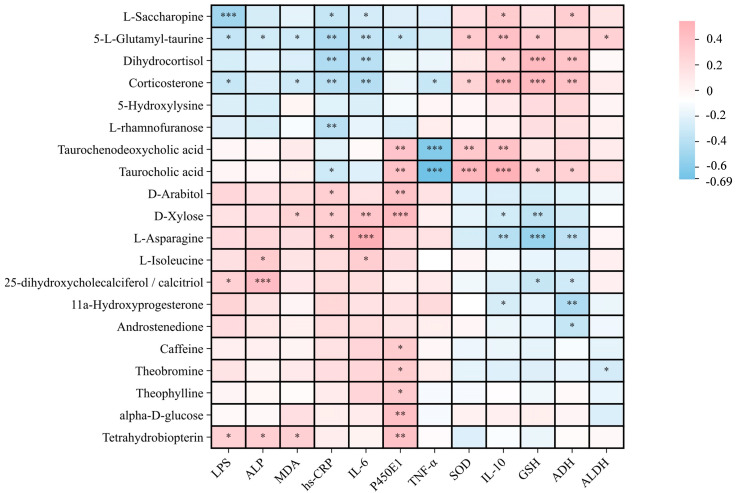
Correlation analysis between key metabolites and clinical measures between the placebo and BC179 groups. (In this figure, the significance levels are marked as follows: * indicates *p* < 0.05, ** indicates *p* < 0.01, and *** indicates *p* < 0.001.)

**Table 1 antioxidants-14-01038-t001:** Baseline characteristics of participants.

Projects	BC179 Group (n = 15)		Placebo Group (n = 15)		
	Mean	SD	Mean	SD	*p*-Value
Age	30.400	13.16	31.93	12.578	0.747
BMI	23.67	2.83	22.81	2.54	0.386
Sex (M/F)	10/5		12/3		0.408

## Data Availability

The original contributions presented in this study are included in the article. Further inquiries can be directed to the corresponding author.

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
