# Peer review of "Weizmannia coagulans BC179 Alleviates Post-Alcohol Discomfort May via Taurine-Related Metabolism and Antioxidant Regulation: A Randomized, Double-Blind, Placebo-Controlled Trial"

_antioxidants, 2025, doi:10.3390/antiox14091038_

Round 1
Reviewer 1 Report
The manuscript is very interesting. However, I have the following comments.
1. The introduction is sufficient, but I suggest the authors include more background related to the hepatic metabolism of alcohol, and how alcohol intake generates hepatic and systemic oxidative stress.
2. Considering the study's objective, the methodology used is adequate, but it would be very beneficial if the authors included a dietary study, considering the intervention time and the possible effects on food, energy, and nutrient intake.
3. Regarding the results, I suggest:
- Improve the resolution of the images; it was not easy for me to read and understand each one.
- In the figures and in the legend for each figure, be very specific regarding the origin of the results. For example: were the oxidative stress parameters obtained from whole blood or from a specific cell?
- Consider increasing the size of the figures (they are very small).
- The results are interesting and could be very useful for other research and clinical interventions. In this regard, I think it would be very important for the authors to highlight the relevance of the damage caused by alcohol to the liver. Currently, the high prevalence of fatty liver disease in adults is caused by a poor-quality diet, but also by high alcohol intake. This intake is generally underreported. Please consider the latest international consensus on this topic. PMID: 39270816
The manuscript is very interesting. However, I have the following comments.
1. The introduction is sufficient, but I suggest the authors include more background related to the hepatic metabolism of alcohol, and how alcohol intake generates hepatic and systemic oxidative stress.
2. Considering the study's objective, the methodology used is adequate, but it would be very beneficial if the authors included a dietary study, considering the intervention time and the possible effects on food, energy, and nutrient intake.
3. Regarding the results, I suggest:
- Improve the resolution of the images; it was not easy for me to read and understand each one.
- In the figures and in the legend for each figure, be very specific regarding the origin of the results. For example: were the oxidative stress parameters obtained from whole blood or from a specific cell?
- Consider increasing the size of the figures (they are very small).
- The results are interesting and could be very useful for other research and clinical interventions. In this regard, I think it would be very important for the authors to highlight the relevance of the damage caused by alcohol to the liver. Currently, the high prevalence of fatty liver disease in adults is caused by a poor-quality diet, but also by high alcohol intake. This intake is generally underreported. Please consider the latest international consensus on this topic. PMID: 39270816
Author Response
For research article
|
Response to Reviewer 1 Comments |
||
|
1. Summary |
|
|
|
Thank you very much for taking the time to review this manuscript. Please find the detailed responses below and the corresponding revisions in the re-submitted files. |
||
|
2. Questions for General Evaluation |
Reviewer’s Evaluation |
Response and Revisions |
|
Does the introduction provide sufficient background and include all relevant references? |
Can be improved |
Please read the following detailed response. |
|
Are all the cited references relevant to the research? |
Yes |
|
|
Is the research design appropriate? |
Yes |
|
|
Are the methods adequately described? |
Can be improved |
Please read the following detailed response. |
|
Are the results clearly presented? |
Can be improved |
Please read the following detailed response. |
|
Are the conclusions supported by the results? |
Yes |
|
|
3. Point-by-point response to Comments and Suggestions for Authors |
||
|
Comments 1: The introduction is sufficient, but I suggest the authors include more background related to the hepatic metabolism of alcohol, and how alcohol intake generates hepatic and systemic oxidative stress. |
||
|
Response 1: Thank you very much for your valuable suggestion. We fully agree with your view that supplementing more background information on the hepatic metabolism of alcohol and the mechanisms by which alcohol intake induces hepatic and systemic oxidative stress will further enhance the comprehensiveness and depth of the introduction.In the revised manuscript, we have added relevant content in the introduction section. See lines 42 to 78 for details |
||
|
Comments 2: Considering the study's objective, the methodology used is adequate, but it would be very beneficial if the authors included a dietary study, considering the intervention time and the possible effects on food, energy, and nutrient intake. |
||
|
Response 2: We fully agree that dietary factors, as an important variable influencing research results, their interaction with intervention measures is indeed worthy of in-depth exploration. In the pre-experiment,In the pre-experiment, we used the same diet as seen in 119-121. In the formal trial, we adopted the recommended daily ration(50% - 60% carbohydrates, 15% - 20% protein, 20% - 30% fat), as shown in lines 144-147. In the subsequent others, we will focus on supplementing the monitoring of dietary data at different stages of intervention. We will collect relevant information through methods such as dietary records and dietary surveys to conduct a more comprehensive analysis of the intervention effect and its potential influencing factors. |
||
|
Comments 3: Regarding the results, I suggest: - Improve the resolution of the images; it was not easy for me to read and understand each one. - In the figures and in the legend for each figure, be very specific regarding the origin of the results. For example: were the oxidative stress parameters obtained from whole blood or from a specific cell? - Consider increasing the size of the figures (they are very small). - The results are interesting and could be very useful for other research and clinical interventions. In this regard, I think it would be very important for the authors to highlight the relevance of the damage caused by alcohol to the liver. Currently, the high prevalence of fatty liver disease in adults is caused by a poor-quality diet, but also by high alcohol intake. This intake is generally under reported. Please consider the latest international consensus on this topic. PMID: 39270816 |
||
|
Response 3: Thank you very much for your valuable suggestion. 1. We have enhanced the resolution of the images. For details, see 264-267, 303-306, 319-321, 339-341. 2. Regarding the details of the charts, we have provided specific explanations and optimizations for the legends in the revised draft. 3. We referred to relevant literature to ensure the scientific and practical nature of the conclusion. For details, see 416-419 |
||
|
4. Response to Comments on the Quality of English Language |
||
|
Point 1: The English is fine and does not require any improvement. |
||
|
Response 1:Nothing |
||
|
5. Additional clarifications |
||
|
Nothing |
||
Reviewer 2 Report
Before further steps, authors need to address the following issues.
- I believed that, based on the shown evidence, in the title of the investigation the implication of taurine-related metabolism was overrated by authors since the taurine -related metabolites were not determined using mass spectrometry grade internal standards to conclusively validate their regulation. So, the title of the research should be modified.
- In result section, authors state the following “Concurrently, Figure 3D demonstrates that after at least 30 days of intervention, the BC179 group showed a significant 0.57 μmol/mL reduction in blood alcohol concentration relative to the placebo group (p < 0.05)”; however, panel 3D of that figure shows the ALP content. Moreover, in result subsection “3.2.3. Serum alkaline phosphatase and intestinal barrier”, authors state the following “From Figure 3E and F, the BC179 group exhibited a significant post-alcohol decrease in ALP levels (p < 0.001), with a 5.55 ng/mL reduction compared to the placebo group”; however, panel 3E of that figure shows BAC. So, authors must carefully correct these issues.
- Panels and fonts of figure 6 are too small. Authors should be aware that figures in a manuscript must be clearly visible to make it easy and fluent for readers; however, that figure does not follow the rule. Authors should split figure 6 in at least three different ones in order to improve its presentation.
- Abbreviations/symbols written in parentheses, after definitions, should only be cited at the first appearance, subsequently, only the abbreviation/symbol should be cited throughout the manuscript. However, authors repeat the definition and abbreviation of several sentences throughout the manuscript. For example, blood alcohol concentration (BAC), is defined and abbreviated in both materials and methods sections; as well as, alcohol dehydrogenase (ADH) and aldehyde dehydrogenase (ALDH), are defined and abbreviated in both introduction and in materials and methods section several times. Some others definition and abbreviations have the same issues. Authors must carefully correct them throughout the manuscript.
- “Figure 1. Flow diagram of the experimental study design” was not cited anywhere in the text before showing the figure.
- Kits and reagents should contain the full vendor information, such as catalogue number, company, place and country of production. Authors should include the missing information in all reagents cited in materials and methods section. For example, that missing in “2.4. Blood sample collection and determination” subsection.
Please, see major comments
Author Response
For research article
|
Response to Reviewer 1 Comments |
||
|
1. Summary |
|
|
|
Thank you very much for taking the time to review this manuscript. Please find the detailed responses below and the corresponding revisions in the re-submitted files. |
||
|
2. Questions for General Evaluation |
Reviewer’s Evaluation |
Response and Revisions |
|
Does the introduction provide sufficient background and include all relevant references? |
Can be improved |
Please read the following detailed response. |
|
Are all the cited references relevant to the research? |
Yes |
|
|
Is the research design appropriate? |
Yes |
|
|
Are the methods adequately described? |
Yes |
|
|
Are the results clearly presented? |
Can be improved |
Please read the following detailed response. |
|
Are the conclusions supported by the results? |
Yes |
|
|
3. Point-by-point response to Comments and Suggestions for Authors |
||
|
Comments 1: I believed that, based on the shown evidence, in the title of the investigation the implication of taurine-related metabolism was overrated by authors since the taurine -related metabolites were not determined using mass spectrometry grade internal standards to conclusively validate their regulation. So, the title of the research should be modified. |
||
|
Response 1: We fully understand your prudent consideration regarding the strength of the experimental evidence regarding the issue that taurine-related metabolites were not ultimately validated using mass spectrometry-grade internal standards. During the research process, we initially identified the changing trends of taurine-related metabolites through untargeted metabolomics and speculated that they might be involved in the process by which BC179 alleviates post-alcohol discomfort. The quantification of metabolites in the full-spectrum metabolism -LCMS metabolome is all based on the relative quantification of peak area. However, the experiment set up X biological replicates and strictly controlled the accuracy of the entire experiment and the validity of the experimental data based on mixed internal standards and quality control samples (QC). Subsequently, the key metabolites in the metabolic pathways will be verified. However, as you have pointed out, due to the lack of absolute quantitative validation using mass spectrometry-grade internal standards, the current evidence is indeed insufficient to fully confirm the specific regulatory amplitude and direct causal relationship of taurine-related metabolism. The expression of its role in the title is indeed not rigorous enough. Based on your suggestions, we have revised the title to "Weizmannia coagulans BC179 alleviates post-alcohol discomfort may via taurine-related metabolism and antioxidant regulation: a randomized, double-blind, placebo-controlled trial". This revision weakens the deterministic expression of the taurine-related metabolic mechanism and more objectively reflects the strength of the existing evidence. |
||
|
Comments 2: In result section, authors state the following “Concurrently, Figure 3D demonstrates that after at least 30 days of intervention, the BC179 group showed a significant 0.57 μmol/mL reduction in blood alcohol concentration relative to the placebo group (p < 0.05)”; however, panel 3D of that figure shows the ALP content. Moreover, in result subsection “3.2.3. Serum alkaline phosphatase and intestinal barrier”, authors state the following “From Figure 3E and F, the BC179 group exhibited a significant post-alcohol decrease in ALP levels (p < 0.001), with a 5.55 ng/mL reduction compared to the placebo group”; however, panel 3E of that figure shows BAC. So, authors must carefully correct these issues. |
||
|
Response 2: Thank you very much for your meticulous review and valuable comments. We have made the revisions. For details, please refer to 303-306 At present, we have systematically reviewed and corrected all content in the full text related to the correspondence between figure numbers and indicators, ensuring that the result descriptions, figure numbers, and displayed indicators are strictly consistent, so as to avoid similar problems from occurring again. The revised content will present the research data more accurately and enhance the rigor of the manuscript. |
||
|
Comments 3: Panels and fonts of figure 6 are too small. Authors should be aware that figures in a manuscript must be clearly visible to make it easy and fluent for readers; however, that figure does not follow the rule. Authors should split figure 6 in at least three different ones in order to improve its presentation. |
||
|
Response 3: To address this issue, we have optimized the figure in accordance with your suggestions: the original Figure 6 has been split into 2 independent subfigures. The revised figures not only meet the journal's requirements for readability but also present the logical relationships of various data sections more intuitively. For details, please refer to 345-347, 377-379. Comments 4:Abbreviations/symbols written in parentheses, after definitions, should only be cited at the first appearance, subsequently, only the abbreviation/symbol should be cited throughout the manuscript. However, authors repeat the definition and abbreviation of several sentences throughout the manuscript. For example, blood alcohol concentration (BAC), is defined and abbreviated in both materials and methods sections; as well as, alcohol dehydrogenase (ADH) and aldehyde dehydrogenase (ALDH), are defined and abbreviated in both introduction and in materials and methods section several times. Some others definition and abbreviations have the same issues. Authors must carefully correct them throughout the manuscript. Response 4:Thank you very much for pointing out the issue regarding the usage standards of abbreviations/symbols. The repeated definition of abbreviations throughout the manuscript that you mentioned, such as BAC,ADH, and ALDH, indeed violates the academic writing standard of "using only the abbreviation after its first definition." This was caused by our lack of rigor during the manuscript writing and proofreading process, and we sincerely apologize for this. To address this issue, we have systematically reviewed all abbreviations and symbols in the full text:We have uniformly provided complete definitions and marked abbreviations for each abbreviation in the section where it first appears (with priority given to the first mention in the Introduction or Materials and Methods), such as blood alcohol concentration, BAC and alcohol dehydrogenase, ADH; For all subsequent occurrences after the first definition, only the abbreviated form is used, with the repeated full names and the abbreviations in parentheses removed;Additionally, we have checked the consistency of all abbreviations in the text to ensure that the abbreviation for the same term is uniform throughout the manuscript (e.g., avoiding different abbreviations for the same indicator). Comments 5:“Figure 1. Flow diagram of the experimental study design” was not cited anywhere in the text before showing the figure. Response 5:Thank you very much for your valuable suggestion. To address this issue, we have added a reference to Figure 1 in the paragraph of the main text describing the experimental study design, clearly stating that "the detailed flow of the experimental study design is shown in Figure 1 (Flowchart of the experimental study design)". For details, please refer to 112, 133-134. Comments 6:Kits and reagents should contain the full vendor information, such as catalogue number, company, place and country of production. Authors should include the missing information in all reagents cited in materials and methods section. For example, that missing in “2.4. Blood sample collection and determination” subsection. Response 6:Thank you very much for your valuable suggestion. In particular, for the detection kits mentioned in key subsections such as "2.4. Blood sample collection and determination" (e.g., alcohol concentration detection kits, antioxidant index determination reagents, etc.), we have fully supplemented the above-mentioned information to ensure that readers can accurately trace the source of experimental materials. For details, please refer to 167-173. |
||
|
4. Response to Comments on the Quality of English Language |
||
|
Point 1: The English is fine and does not require any improvement. |
||
|
Response 1:Nothing |
||
|
5. Additional clarifications |
||
|
Nothing |
||

Reviewer 3 Report
The study provides a detailed analysis of how probiotic bacterial strains could potentially be used to reduce alcohol toxicity in humans and has some very relevant and interesting studies related to this concept.
In the manuscript entitled “Weizmannia coagulans BC179 alleviates post-alcohol discomfort via taurine-related metabolism and antioxidant regulation:a randomized, double-blind, placebo-controlled trial” the authors have examined the role of the probiotic strain, Weizmannia coagulans 19 or BC179 in controlling post alcohol discomfort in patients compared to placebo controls. The studies show some significant effects of this probiotic on toxicity related parameters caused by alcohol such as CYP2E1, inflammatory markers and alcohol metabolizing enzymes. Overall, the study is well-designed. A few concerns noted are:
- The authors measure activity of ADH and ALDH at pre-experimental time points and show that BC179 increases their activity. What about during hangover times (45 d) as studied in figure 3? Are both active forms of these enzymes and their total level altered by BC179 at these times?
- The methods section should specify how the ADH and ALDH activities are measured by the authors.
- Since taurine is known to increase activity of alcohol dehydrogenase or ADH, which is seen by the authors in pre-experimental conditions, a discussion of a potential role of taurine in facilitating BC179 dependent ADH activity should be discussed with appropriate references.
- In figure 6, the results should clearly specify which metabolites related to taurine metabolism are regulated after BC179 exposure. Or is it that taurine levels go up? The figure and result are not clear about this. Also, the heatmap is so small that it is not possible to read it even at higher magnification.
Author Response
For research article
|
Response to Reviewer 2 Comments |
||
|
1. Summary |
|
|
|
Thank you very much for taking the time to review this manuscript. Please find the detailed responses below and the corresponding revisions in the re-submitted files. |
||
|
2. Questions for General Evaluation |
Reviewer’s Evaluation |
Response and Revisions |
|
Does the introduction provide sufficient background and include all relevant references? |
Yes |
|
|
Are all the cited references relevant to the research? |
Yes |
|
|
Is the research design appropriate? |
Yes |
|
|
Are the methods adequately described? |
Can be improved |
Please read the following detailed response. |
|
Are the results clearly presented? |
Yes |
|
|
Are the conclusions supported by the results? |
Yes |
|
|
3. Point-by-point response to Comments and Suggestions for Authors |
||
|
Comments 1: The authors measure activity of ADH and ALDH at pre-experimental time points and show that BC179 increases their activity. What about during hangover times (45 d) as studied in figure 3? Are both active forms of these enzymes and their total level altered by BC179 at these times? |
||
|
Response 1: Figure 2 presents the results of the preliminary experiment, showing the activities of ADH and ALDH at different time points within 240 minutes after alcohol consumption. Since the core of the study is that BC179 alleviates post-alcohol discomfort by regulating metabolism and oxidative stress, the formal experimental results in Figure 3 do not investigate the levels of these enzymes at different time points. However, the overall activity levels of these enzymes have increased. |
||
|
Comments 2: The methods section should specify how the ADH and ALDH activities are measured by the authors. |
||
|
Response 2: Thank you for pointing this out. We agree with this comment. How to measure ADH and ALDH activities has been explained in 125-131 and 167-173. |
||
|
Comments 3: Since taurine is known to increase activity of alcohol dehydrogenase or ADH, which is seen by the authors in pre-experimental conditions, a discussion of a potential role of taurine in facilitating BC179 dependent ADH activity should be discussed with appropriate references. |
||
|
Response 3: Thank you for your valuable suggestions. We have included three references in the manuscript. This hint helps us to explore more deeply the potential mechanism by which BC179 alleviates post-alcohol discomfort. The results of this study show that after BC179 intervention, the taurine metabolic pathway is significantly enhanced, and the ADH activity increases simultaneously. In light of the known research background that taurine upregates alcohol dehydrogenase (ADH) activity, in-depth discussions were conducted in combination with relevant literature. For details, please refer to 515-528, 633-638. Comments 4:In figure 6, the results should clearly specify which metabolites related to taurine metabolism are regulated after BC179 exposure. Or is it that taurine levels go up? The figure and result are not clear about this. Also, the heat map is so small that it is not possible to read it even at higher magnification. Response 4: Thank you very much for your valuable suggestion. Regarding the regulation of taurine metabolism-related metabolites in Figure 6: We have supplemented specific information in the Results section, clearly listing the taurine metabolism-related metabolites that changed significantly after BC179 exposure, as shown in lines 373-375. Regarding the issue of the heat map being too small: We have recreated the heat map in Figure 6, which is now presented as Figure 7. The image size and font clarity have been increased to ensure that the names of each metabolite and the changes in their expression levels can be clearly identified during regular reading and after magnification. |
||
|
4. Response to Comments on the Quality of English Language |
||
|
Point 1: The English is fine and does not require any improvement. |
||
|
Response 1:Nothing |
||
|
5. Additional clarifications |
||
|
Nothing |
||

Round 2
Reviewer 1 Report
The authors answered all comments. I don't have more questions and inquiry. Therefore, the manuscript can be accepted.
The authors answered all comments. I don't have more questions and inquiry. Therefore, the manuscript can be accepted.
Reviewer 2 Report
The authors have properly addressed all my suggestions.
The authors have properly addressed all my suggestions.
Reviewer 3 Report
Revision concerns are addressed
Revision concerns are addressed.